METHODS AND RESOURCES

# Assessing the replicability of spatial gene expression using atlas data from the adult mouse brain

**Shaina Lu**[1], **Cantin Ortiz**[2], **Daniel Fürth**[1], **Stephan Fischer**[1], **Konstantinos Meletis**[2], **Anthony Zador**[1]*, **Jesse Gillis**[1]*

**1** Cold Spring Harbor Laboratory, Cold Spring Harbor, New York, United States of America, **2** Department of Neuroscience, Karolinska Institutet, Solna, Sweden

* zador@cshl.edu (AZ); jgillis@cshl.edu (JG)

**Data Availability Statement:** The ABA data is available publicly for download directly from the Allen Brain Atlas website (http://help.brain-map.org/display/api/Allen%2BBrain%2BAtlas%2BAPI). ST data (from Ortiz et al., 2020) is available at

## Abstract

High-throughput, spatially resolved gene expression techniques are poised to be transformative across biology by overcoming a central limitation in single-cell biology: the lack of information on relationships that organize the cells into the functional groupings characteristic of tissues in complex multicellular organisms. Spatial expression is particularly interesting in the mammalian brain, which has a highly defined structure, strong spatial constraint in its organization, and detailed multimodal phenotypes for cells and ensembles of cells that can be linked to mesoscale properties such as projection patterns, and from there, to circuits generating behavior. However, as with any type of expression data, cross-dataset benchmarking of spatial data is a crucial first step. Here, we assess the replicability, with reference to canonical brain subdivisions, between the Allen Institute's in situ hybridization data from the adult mouse brain (Allen Brain Atlas (ABA)) and a similar dataset collected using spatial transcriptomics (ST). With the advent of tractable spatial techniques, for the first time, we are able to benchmark the Allen Institute's whole-brain, whole-transcriptome spatial expression dataset with a second independent dataset that similarly spans the whole brain and transcriptome. We use regularized linear regression (LASSO), linear regression, and correlation-based feature selection in a supervised learning framework to classify expression samples relative to their assayed location. We show that Allen Reference Atlas labels are classifiable using transcription in both data sets, but that performance is higher in the ABA than in ST. Furthermore, models trained in one dataset and tested in the opposite dataset do not reproduce classification performance bidirectionally. While an identifying expression profile can be found for a given brain area, it does not generalize to the opposite dataset. In general, we found that canonical brain area labels are classifiable in gene expression space within dataset and that our observed performance is not merely reflecting physical distance in the brain. However, we also show that cross-platform classification is not robust. Emerging spatial datasets from the mouse brain will allow further characterization of cross-dataset replicability ultimately providing a valuable reference set for understanding the cell biology of the brain.

molecularatlas.org and GEO (https://www.ncbi.
nlm.nih.gov/geo/query/acc.cgi?acc=GSE147747).

**Funding:** SL is supported by the Edward and
Martha Gerry Fellowship funded by The William
Stamps Farish Fund and the Gladys and Roland
Harriman Foundation. DF is supported by NARSAD
Young Investigator Grant from Brain & Behavior
Research Foundation. SF is supported by NIH
Grant U19MH114821. KM is supported by a grant
from the Swedish Research Council (VR project
2018-00608). JG is supported by NIH Grants
R01MH113005 and R01LM012736. AZ is
supported by NIH Grants 5RO1NS073129,
5RO1DA036913, RF1MH114132, and
U01MH109113, the Brain Research Foundation
(BRF-SIA-2014-03), IARPA MICrONS
[D16PC0008], Paul Allen Distinguished
Investigator Award, Chan Zuckerberg Initiative
(2017-0530 ZADOR/ALLEN INST(SVCF) SUB
awarded to A.M.Z], and Robert Lourie. The funders
had no role in study design, data collection and
analysis, decision to publish, or preparation of the
manuscript.

**Competing interests:** I have read the journal's
policy and the authors of this manuscript have the
following competing interests: AZ is a founder and
equity owner of Cajal Neuroscience and a member
of its scientific advisory board.

**Abbreviations:** ABA, Allen Brain Atlas; ARA, Allen
Reference Atlas; AUROC, area under the receiver
operating curve; CFS, correlation-based feature
selection; DE, differential expression; ISH, in situ
hybridization; k-NN, k-nearest neighbors; LASSO,
least absolute shrinkage and selection operator;
MWU, Mann–Whitney $U$; PCA, principal
component analysis; ST, spatial transcriptomics.

# Background

In the last 5 years, there has been an explosion of spatially resolved transcriptomics techniques that have made it possible to easily sequence whole transcriptomes while retaining fine-scale spatial information [1–5]. These new technologies are poised to be transformative across biology [6]. Despite the recent proliferation and improvement of single-cell technologies, these technologies largely depend on tissue dissociation and thus lack information on the spatial origin of sequenced cells. New spatial sequencing tools fill this gap, allowing us to understand the spatial patterning of cell-type specific expression. The stereotyped spatial organization and transcriptional heterogeneity of the brain make it an especially appealing application of these new technologies. Spatial gene expression has the potential to serve as a link between the molecular, mesoscale, and emergent properties of the brain such as gene expression, circuitry, and behavior, respectively [7,8]. This, in turn, could lead to tackling long-standing questions about the brain, such as how gene expression relates to connectivity of neurons or how spatial patterning of expression drives development. Emerging experimental approaches [9–11] and techniques [12–16] have already begun to link multisource information from the mouse brain. However, in order to perform robust multimodality studies, we must first assess replicability within one type of data. Given the potential of spatial transcriptomics (ST) approaches in neuroscience, the early availability of spatial data, and the stereotyped substructure, we use the adult mouse brain as a model system for a cross-platform characterization of spatial data.

Over a decade ago, the first whole-transcriptome, spatially resolved gene expression dataset from the adult mouse brain was collected by the Allen Institute using in situ hybridization (ISH) (Allen Brain Atlas (ABA)) [17,18]. Since its release, this dataset has become a cornerstone for modern neurobiologists who often use it as a first point of reference for gene expression in the mouse brain. The generation of this dataset was a laborious effort requiring many years, the work of many scientists, and many sacrificed mice. The influx of technologies preserving the spatial origin of transcripts presents the opportunity to assess the generalizability of the ABA data for the first time. As the sole reference spatial dataset, benchmarking the ABA data is essential to assess the robustness of the observed gene expression patterns across distinct experiments and technological platforms. In this manuscript, we use "benchmarking" to refer to the assessment of replicability across independent datasets representing different experimental techniques. Obtaining replicable results across gene expression assays is notoriously challenging, so cross-platform, cross-dataset transcriptomics benchmarking has proved crucial since early transcriptome assays in the form of microarrays [19,20].

To address this need for ST and cross-modality robustness in the brain, here we undertook a whole-brain benchmarking of the ABA via linking gene expression and anatomy. We analyzed a spatial gene expression dataset from one adult mouse brain collected using ST [21] (see Methods) alongside the ABA. ST is a spatially barcoded mRNA capture technique followed by sequencing readout, while the ABA dataset is a collection of single-molecule ISH experiments across the whole transcriptome [1,17]. While benchmarking of the 2 datasets could be done on many scales, we chose to look across brains and across techniques with reference to named brain areas. This approach contains noise associated with the relative biases of each technique (different assays); experimental noise from tissue processing and alignment; biological variability (different brains); and variability from brain area segmentation and naming itself. Despite all these potential sources of noise, our approach combining spatial gene expression with brain area identity allows us to focus on biological conclusions that could be drawn from replicable spatial data. Not readily available with more technical approaches to benchmarking, our approach allowed us to pursue a biological question. We principally ask if canonical, anatomically defined brain areas from the Allen Reference Atlas (ARA) can be assigned using

gene expression alone and, in corollary, how well these assignments replicate across the ABA and ST datasets. We use an interpretable supervised learning framework for classification, where the target values are the ARA brain area labels and the features are the gene expression profiles for samples from across the whole brain (Fig 1A and 1B). We choose to use linear modeling to maintain easily interpretable models that can be related to underlying biology.

Using this approach, we show that ARA labels are classifiable using gene expression, but that performance is higher in the ABA than in ST. We further demonstrate that models trained in one dataset and tested in the opposite dataset do not reproduce classification performance bidirectionally. We then identify potential biological explanations for the difference in cross-dataset performance in classifying brain areas. Finally, we found that although an identifying gene expression profile can always be found for a given brain area, it does not generalize to the opposite dataset. In summary, within each dataset, canonical brain area labels were classifiable and meaningful in gene expression space, but replicability across these 2 very different assays of gene expression was not robust.

## Results and discussion

### Allen Reference Atlas brain areas are classifiable using gene expression alone

With the advent of new high-throughput capture technologies for ST, we present, as is necessary for all new biological assays, a cross-technology assessment of generalizability in a well-characterized model system: the adult mouse brain. These new technologies allow, for the first time, the cross-platform assessment of canonical, atlas brain area subdivisions relative to gene expression at a whole-brain scale. Traditionally, parcellation of the mouse brain has depended on anatomical landmarks and cytoarchitecture, at times, including interregion connectivity and molecular properties [17,22,23]. By enabling the relatively rapid and high-throughput collection of spatially resolved, whole-transcriptome data in the adult mouse brain, these new spatial assays pave the way for a multimodality assessment of canonical brain area labels. Specifically, in the present work, we ask if brain areas from the ARA [17] are classifiable using 2 spatial gene expression datasets: the Allen Institute's own ISH data [17,18] and a second dataset collected using ST [1,21] (Fig 1A and 1B). After filtering, the ABA consists of 62,527 voxels (rows) with expression from 19,934 unique genes (columns) mapping to 569 nonoverlapping brain area labels, and the ST consists of 30,780 spots (rows) with 16,557 genes (columns) mapping to 461 brain area labels (see Methods for details). The ABA dataset consists of a minimum of roughly 3,260 brains, while the ST dataset is collected from 3 mice (17,21) (see Methods). Comparing accuracy in classification of ARA brain areas across 2 technological platforms and datasets allows us to draw conclusions about spatial expression that are more likely to be biological and generalizable than subject to the technical biases of any one dataset.

To determine if we could more generally determine canonical brain areas from spatial gene expression, we first asked if we could do so within each of the 2 datasets independently. Given the known high correlation structure of gene expression [24], we hypothesized that we could determine the brain area of origin of a gene expression sample using only a subset of the total genes. Fitting these criteria, we chose least absolute shrinkage and selection operator, or LASSO regression [25]. LASSO is a regularized linear regression model that minimizes the L1 norm of the coefficients (i.e., the sum of the absolute values of the coefficients). LASSO typically drives most coefficients toward zero and thus leaves few genes contributing to the final model; LASSO in effect picks "marker genes" of spatial expression in the brain. We use LASSO in a supervised learning framework with a random 50/50 train–test split for two-class classification of all pairwise brain areas successively (Fig 1C) (see Methods). The brain areas included

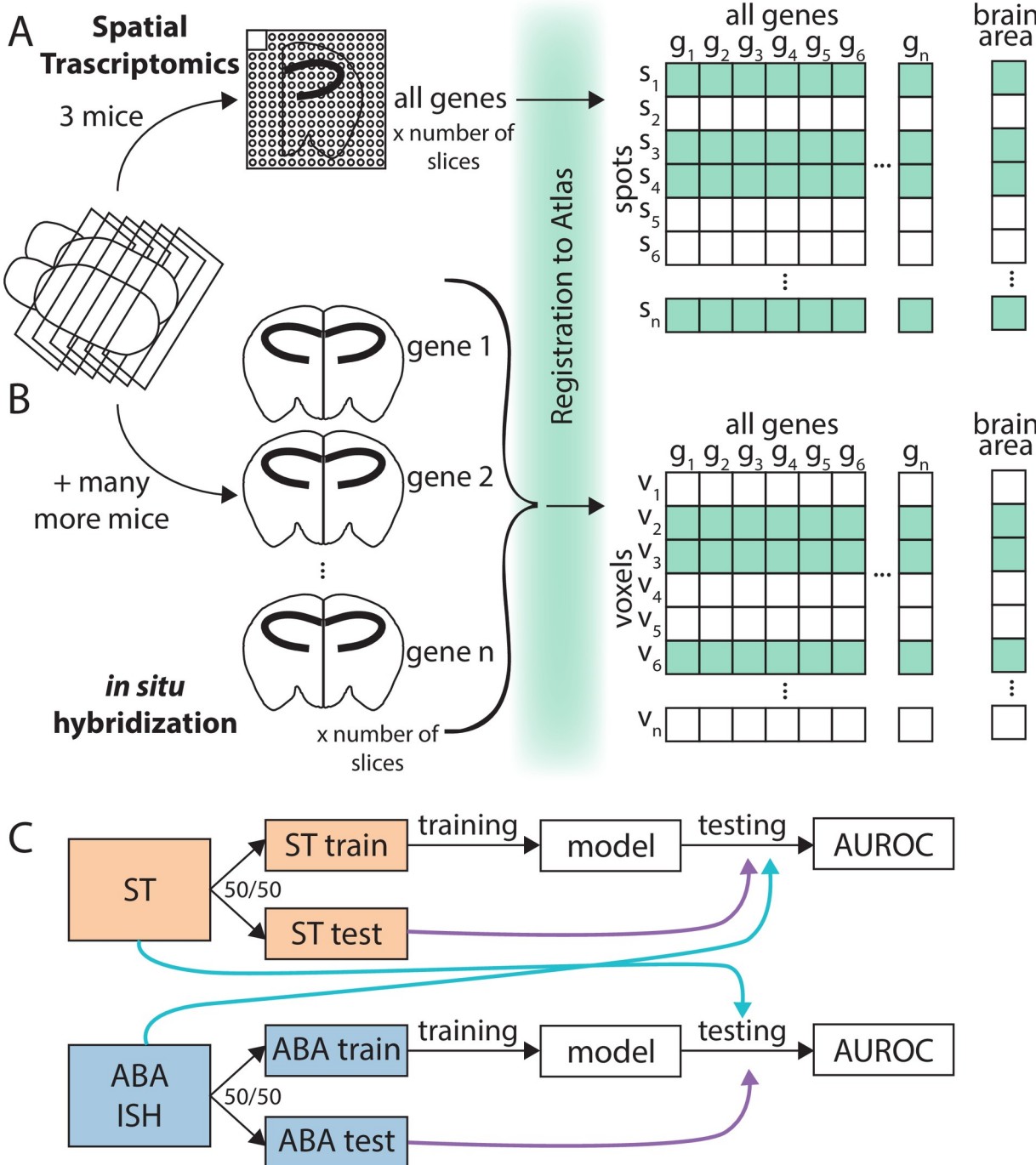

**Fig 1. Collection and processing of spatial gene expression datasets.** (A) Schematic depicting workflow of collecting whole-brain spatial gene expression using ST. Illustration depicts sectioning of mouse brain, tissue from one hemisphere on one ST slide, registration to ARA, and a layout of the collected data. (B) Schematic depicting workflow of collecting Allen Institute's whole-brain spatial gene expression using ISH (ABA). Illustration depicts similar workflow to (A), but instead of ST capturing all genes in one (3 for this dataset) brain, there were many more mice used to collect the whole-transcriptome dataset since each brain tissue slice can only be used to probe one gene. (C) Schematic illustrating classification schema. The ST dataset from (A) (orange) and ABA dataset form (B) (blue) were split into 50/50 train/test folds. The training fold was used for model building and the test fold for evaluating the trained model within dataset (purple arrow). Later analysis also applied models trained using the train fold of one dataset to the opposite dataset for testing (light blue arrow). ABA, Allen Brain Atlas; ARA, Allen Reference Atlas; AUROC, area under the receiver operating curve; ISH, in situ hybridization; ST, spatial transcriptomics.

here are nonoverlapping and are the smallest brain areas present in the ARA naming hierarchy. We subsequently refer to these areas as leaf brain areas since they form the leaves of the tree-based representation of the ARA-named brain areas [17]. The performance of the test set classification is reported using the area under the receiver operating curve (AUROC). The AUROC can be thought of as the probability of correctly predicting a given brain region from its gene expression in a comparison with an outgroup (here, a different brain region) and is calculated by taking the predictions from the trained LASSO model and evaluating their correspondence with the known labels in the test fold (see Methods). For example, if ranking the samples by the LASSO predictions separates the samples from the 2 classes perfectly without being interspersed, we would get perfect classification with an AUROC of 1, while a score of 0.5 is random. More generally, in this manuscript, we say a brain area pair is classifiable with respect to each other to indicate a high performance in classification with an AUROC greater than 0.5 and generally closer to 1.

After preliminary filtering (see Methods), we use this approach in both the ST and ABA to classify all the leaf brain areas against each of the others (461 ST areas; 560 ABA areas) (Fig 1C; see Methods). ARA leaf brain areas are classifiable using LASSO (lambda = 0.1) from all other leaf brain areas using only gene expression data from (1) the ABA (mean AUROC = 0.996) (Fig 2A, S1A Fig) and from (2) the ST (mean AUROC = 0.883) (Fig 2B, S1B Fig). These results are consistent across an additional, independent train/test fold split for both datasets (ABA mean AUROC = 0.996, correlation to first split, rho = 0.732; ST mean AUROC = 0.882, correlation to first split, rho = 0.860) (S1C–S1F Fig). As expected, performance falls to chance when brain area labels are permuted as a control (ABA mean AUROC = 0.510; ST mean AUROC = 0.501) (S2A–S2D Fig). Together, these results indicate that there is a set of genes whose expression level can be used to identify it and suggests that canonical brain area labels do reflect spatial patterning of gene expression assayed in both the ABA and ST datasets.

Since our task can be conceived as a multiclass classification problem, we asked if brain area classification performance could be improved using a true multiclass classifier. To test this question, we used the k-nearest neighbors (k-NN) algorithm, which simply assigns the class identity of a test sample based on the majority class label (brain area) of its k closest neighbors in feature (here, expression) space. Using k-NN (k = 5), classification of leaf brain areas fell in ABA (mean AUROC = 0.695; S2E Fig) and ST (mean AUROC = 0.508; S2F Fig) (see Methods). Given the lack of increase in performance and the preferability of our biologically interpretable approach, we choose to continue most analyses using LASSO.

We next asked if single-gene marker selection strategies could outperform LASSO. Highlighting specific brain areas where such markers are known, we looked at classifying the CA2 of the hippocampus and arcuate hypothalamic nucleus with Amigo2 and Pomc, respectively [26–28]. Following long-standing anatomical divisions of the mouse brain, the hippocampal subregions were redefined in the mid-2000s using differences in gene expression [29,30]. Follow-up to the early redefinitions found that while not exclusively expressed in the CA2, Amigo 2 showed high expression levels in the CA2 [28]. Indeed, in the CA2 of the hippocampus, Amigo2 performs better than any other single gene in the ABA (Amigo2 ABA AUROC = 0.920) and ST datasets (Amigo2 ST AUROC = 0.612) (S3A Fig). However, classification of the CA2 using Amigo 2 is still outperformed by the average performance of genes selected by LASSO. One of the major neuronal populations of the arcuate hypothalamic nucleus are the POMC-expressing neurons, shown to have a role in food intake and metabolism [27]. In the arcuate hypothalamic nucleus, Pomc performance in the ABA (Pomc ABA AUROC = 0.993) and ST (Pomc ST AUROC = 0.910) is better than most other single genes and comparable or less than the average LASSO performance for each dataset (S3B Fig). Given

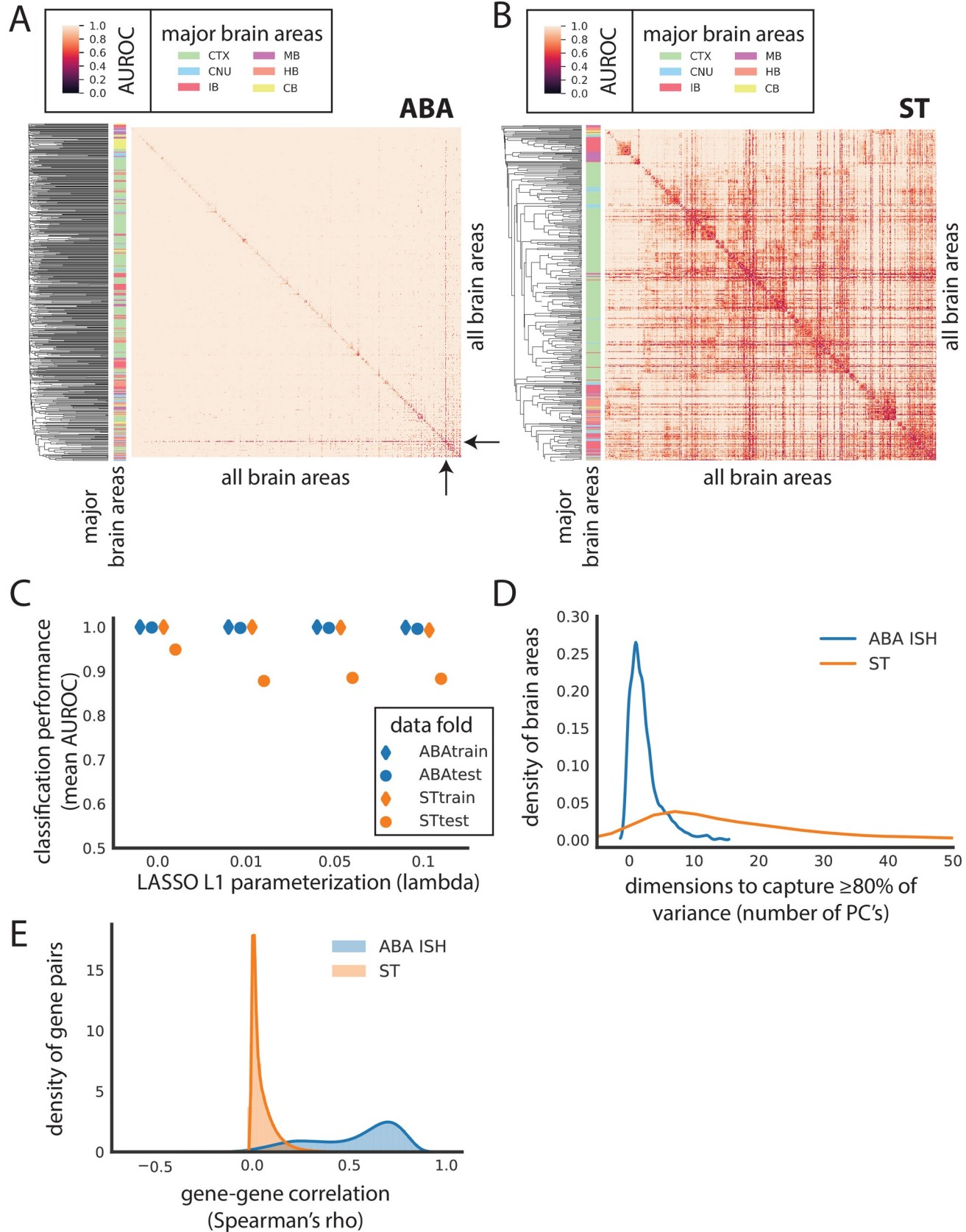

Fig 2. Canonical brain areas are classifiable using gene expression alone in the ABA and ST datasets. Heat map of AUROC for classifying leaf brain areas from all other leaf brain areas in (A) ABA and (B) ST using LASSO (lambda = 0.1). Dendrograms on the far left side represent clustering of leaf brain areas based on the inverse of AUROC; areas with an AUROC near 0.5 get clustered together, while areas with an AUROC near 1 are further apart. Color bar on the left represents the major brain structure that the leaf brain area is grouped under. These areas include CTX, MB, CB, CNU, HB, and IB. (C) Average AUROC (y-axis) of classifying all brain areas from all other brain areas using LASSO across various values of lambda (x-axis): 0, 0.01, 0.05, and 0.1 for ABA train (blue diamond), ABA test (blue dot), ST train (orange diamond), and ST test (orange dot). (D) Number of principal components to capture at least 80% of variance of genes in each of the leaf brain areas after applying PCA to ABA (blue) and ST (orange). ABA brain areas that are larger than ST are randomly down-sampled to have the same number of samples as ST prior to applying PCA. (E) Gene–gene correlations calculated as Spearman's rho between all pairwise genes across the whole dataset for both the ABA (blue) and ST (orange) independently. ABA, Allen Brain Atlas; AUROC, area under the receiver operating curve; CB, cerebellum; CNU, striatum and pallidum; CTX, cortex; HB, hindbrain; IB, thalamus and hypothalamus; ISH, in situ hybridization; LASSO, least absolute shrinkage and selection operator; MB, midbrain; PCA, principal component analysis; ST, spatial transcriptomics.

the comparable performance and, more importantly, since there are not such known markers for most brain areas, we again turned our attention to using LASSO for classifying brain areas.

Notably, performance using LASSO in the ABA is nearly perfect. That the classification in the ABA performs so well is striking, especially considering the potential loss of ISH-level resolution in the voxel representation of the ABA. For the median-performing pair of brain areas in ABA (median AUROC = 1), there is a threshold in classification that can be drawn where all instances of one class can be correctly predicted without any false positives (precision = 1). In contrast, in the ST, no such threshold can be found for the median-performing (median AUROC = 0.959) brain areas (average precision = 0.846) (see Methods). Further, performance in the ABA is consistently higher than the ST across various parameterizations of LASSO (Fig 2C) (see Methods). Despite the comparatively lower performance in the ST, clustering brain areas by AUROC shows brain areas belonging to the same major anatomical region grouping together (Fig 2B) (see Methods). For example, most brain areas belonging to the cortex group together in the middle of the heat map (green bar on left) with a few interspersed areas. This grouping suggests that patterns of expression track with broad anatomical labels. Examining the relative expression of genes that are assayed in both datasets, we see that ranked mean expression is comparable across the 2 datasets (Spearman's ρ = 0.599) (S3C Fig), suggesting that the observed difference in performance is not due to poorly detected genes being well detected in the opposite dataset or vice versa.

Observing the nearly perfect performance in the ABA, we next hypothesized that this dataset may be more low dimensional than suggested by its feature size and may contain many highly correlated features when compared to the ST dataset. We applied principal component analysis (PCA) in each brain area separately by subsetting the data by brain areas, then calculating PCA in each of these subsets independently. Using this approach, we find that on average in individual brain areas, 2 PCs are enough to summarize 80% of the variance per brain area in ABA versus 21 PCs in ST (Fig 2D, S3D Fig) (see Methods). In other words, within each brain area in the ABA, many genes are highly coexpressed. Zooming out to the whole brain, using 200 PCs captures nearly 70% of the variance in ABA compared to nearly 20% in ST (S3E Fig). Further, gene–gene coexpression across the whole dataset is on average higher in the ABA (gene–gene mean Spearman's rho = 0.525) than in the ST (gene–gene mean Spearman's rho = 0.049) (Fig 2E). The perfect performance, low dimensionality on a per brain area basis, and high coexpression all support the idea that although there is meaningful variation in the ABA, it can be captured in few dimensions. In summary, canonical ARA brain areas are classifiable from each other using gene expression alone, but performance is likely inflated in the ABA.

An aside of note is that in the ABA, the one brain area that is consistently lower performing when classified against most other brain areas is the Caudoputamen (mean AUROC = 0.784) (Fig 2A, black arrows). In the ST, the Caudoputamen is not the lowest performing area, but

also has a low mean AUROC (AUROC = 0.619) relative to the other brain areas in ST. In both datasets, the Caudoputamen is the largest leaf brain area composed of the most samples (ABA CP number of voxels = 3,012 versus an average of 85.6 voxels; ST number of spots = 2,051 versus an average of 57 spots). The Caudoputamen is similarly large in other rodent brain atlases, reflecting its lack of cytoarchitectural features [31]. We hypothesized that its relatively larger size could mean that it consists of transcriptomically disparate subsections that are not captured with canonical ARA labeling. Although not an outlier, we do observe that the mean sample correlation for the Caudoputamen in both the ST (mean Pearson's r = 0.727) and ABA (mean Pearson's r = 0.665) is slightly lower than the mean in either case (ST mean Pearson's r = 0.783; ABA mean Pearson's r = 0.696) (S4A Fig). More generally, however, we observe that there is no relationship between size and performance across brain regions (S4B and S4C Fig). In addition to being an outlier in terms of size, the Caudoputamen is the dorsal part of the striatum that encompasses many different functional subdivisions evident through the various corticostriatal projections [31]. Together with the low classification performance of the Caudoputamen using gene expression, this reflects the shortcomings of the ARA Caudoputamen label and the likely need to subdivide the Caudoputamen functionally.

## Cross-dataset learning of Allen Reference Atlas brain areas

**Cross-dataset performance is not bidirectional.** Given the low dimensionality and the near-perfect brain area classification performance in the ABA relative to the ST dataset, we hypothesized that the performance of the LASSO models was artificially inflated in the ABA. To explore this hypothesis, we characterized whether LASSO models trained in one dataset would generalize to the opposite dataset (Fig 1C, light blue arrows). For this step, we further filtered for (1) 445 leaf brain areas that were represented with a minimum of 5 samples in each dataset and for (2) 14,299 overlapping genes (see Methods). In this section, we filtered within-dataset analyses to match this set of genes and leaf areas to maintain a parallel evaluation. LASSO-regularized linear models (lambda = 0.1) trained on ST had a similar within-dataset performance (held-out test fold, mean AUROC = 0.884) and cross-dataset performance (ABA, mean AUROC = 0.829) (Fig 3A and 3B), but the reverse is not true. The performance in classifying pairwise leaf brain areas using LASSO models trained in the ABA (held-out test fold, mean AUROC = 0.997) falls when testing in the ST (mean AUROC = 0.725) (Fig 3A and 3C). These results are consistent across an additional random train/test split for both (1) ST (within-dataset ST test mean AUROC = 0.884; correlation to first split, rho = 0.735) to ABA (ST to ABA cross-dataset mean AUROC = 0.831; correlation to first split, rho = 0.718) (S5A and S5B Fig) and (2) for ABA (within-dataset ABA test mean AUROC = 0.997; correlation to first split, rho = 0.780) to ST (ABA to ST cross-dataset mean AUROC = 0.722; correlation to first split, rho = 0.816) (S5A and S5C Fig). These results show that the ST dataset is more generalizable to the opposite dataset than the ABA. Additionally, this discrepancy in cross-dataset performance suggests that the high performance within the ABA is driven by a property of that dataset not present in the ST (see Discussion).

Given this difference in cross-dataset performance, we next explored if correcting for batch effects improves cross-dataset classification performance. We treated each of the 2 datasets as a batch. Batches within each dataset are not clear, particularly in the ABA where batches might arise independently for each gene, which are sampled as an individual experiment by design of single-molecule ISH. After batch correction between the datasets (see Methods), there is virtually no difference in the mean AUROC for either cross-dataset comparison (ABA held-out test fold mean AUROC = 0.997; ABA to ST mean AUROC = 0.725; ST held-out test fold mean AUROC = 0.884; ST to ABA mean AUROC = 0.829). Looking at individual brain area pairs,

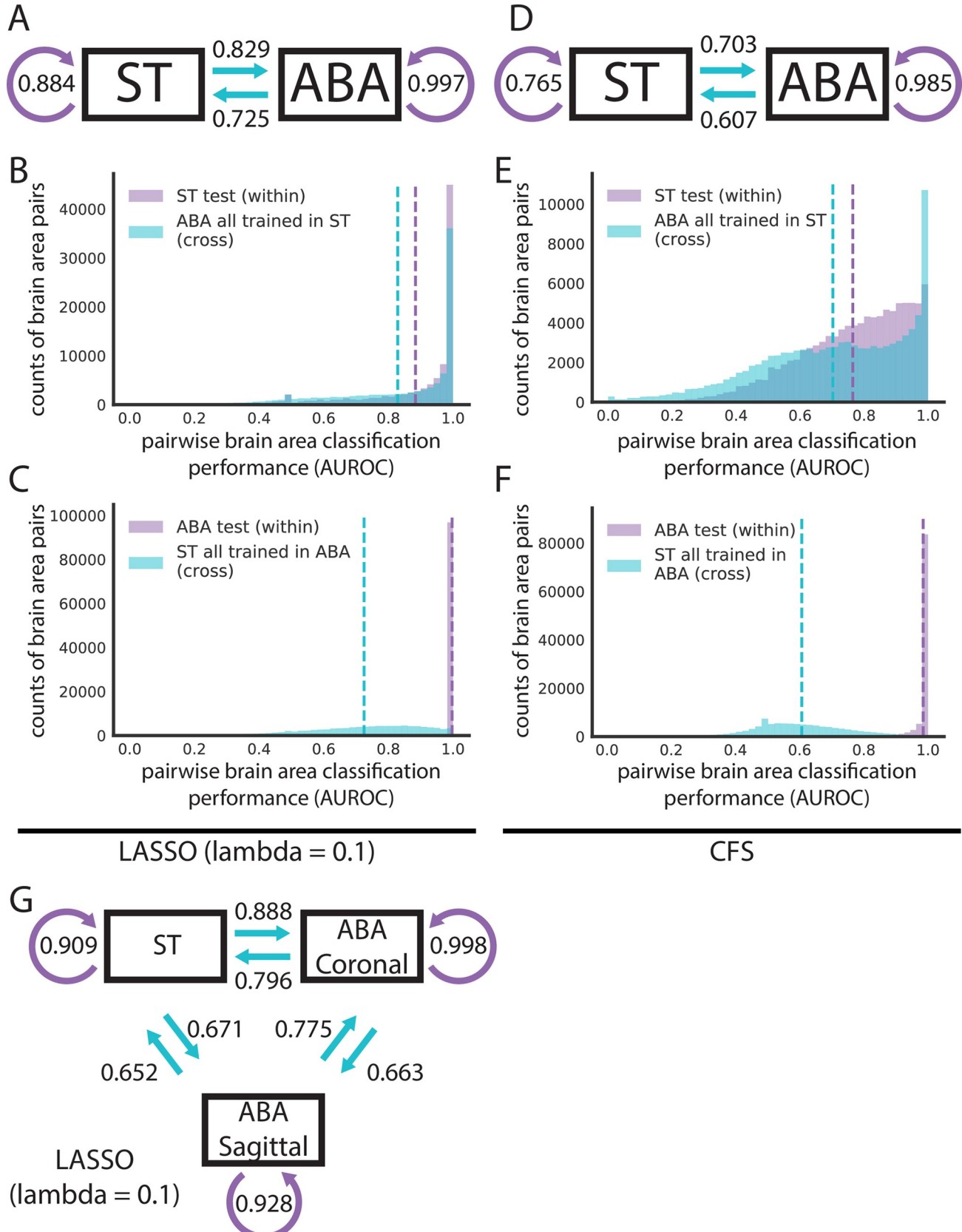

**Fig 3. Cross-dataset learning shows that models do not generalize bidirectionally.** (A and D) Models trained with overlapping genes and brain areas between ST and ABA datasets are evaluated within dataset on the test fold and across dataset on the entire opposite dataset as illustrated in Fig 1C. Summary diagrams showing mean AUROC for within-dataset test set performance (purple arrow) and cross-dataset performance with models trained in the opposite dataset (light blue arrow) for (A) LASSO (lambda = 0.1) and (D) CFS. Distributions of AUROCs for within- (purple) and cross-dataset (light blue) performance for (B) LASSO (lambda = 0.1) trained in ST, (C) LASSO (lambda = 0.1) trained in ABA, (E) CFS trained in ST, and (F) CFS trained in ABA. In all 4 plots, dashed vertical lines represent the mean of the corresponding colored distribution. (G) Summary diagram showing mean AUROCs using LASSO (lambda = 0.1) for separating out the 2 planes of slicing in the ABA and treating them alongside the ST dataset as 3 different datasets for cross-dataset learning. In all 3 summary diagrams (A, D, G), cross-dataset arrows originate from the dataset that the model is trained in and point to the dataset that those models are tested in. ABA, Allen Brain Atlas; AUROC, area under the receiver operating curve; CFS, correlation-based feature selection; LASSO, least absolute shrinkage and selection operator; ST, spatial transcriptomics.

there are some minor differences between uncorrected and corrected classification performance with the largest being for the ST within-dataset held out-test fold (mean absolute difference between corrected and uncorrected = 0.001). Hypothesizing that the much-larger ABA dataset could be driving the batch correction and thus showing very little difference between corrected and uncorrected performance, we down-sampled the ABA to have the same sample size as the ST. Filtering, as before, after down-sampling left us with 414 brain areas. Compared to uncorrected performance when filtering for the same brain areas, there are very small differences in the mean AUROCs (see S1 Table). Visualizing the 2 datasets in principal component space suggests that batch correction may not have much effect since there are no obvious global differences relative to one another (S6A–S6F Fig).

We next ask if the high performance seen within ABA that is lost when models built in the ABA are evaluated in the ST is specific to the LASSO method or a more general feature of the data. To assess the data more directly, we used a second simpler method, correlation-based feature selection (CFS). CFS eliminates model building and simply picks features (genes) that are uncorrelated [32] (see Methods). In this way, CFS parallels LASSO, which implicitly picks uncorrelated feature sets when minimizing its L1 regularized cost function that penalizes additional features.

Using CFS, we picked 100 randomly seeded feature sets for pairwise comparisons of leaf brain areas (see Methods; S7A and S7B Fig). We then took the single best-performing feature set from the train set and evaluated its performance on both the held-out test set and cross-dataset. We did this in both directions, training on both ST and ABA as with LASSO above. CFS can accurately classify pairwise leaf brain areas in both the ST (test set mean AUROC = 0.765) and the ABA (test set mean AUROC = 0.985) (Fig 3D–3F). As with LASSO, classification in ABA with CFS is on average better performing than in ST. Again, following a similar trend as LASSO, the difference in mean cross-dataset performance going from the ST test set to the ABA (difference in mean AUROC = 0.052; mean ST to ABA cross-dataset AUROC = 0.703) is smaller than the reverse (difference in mean AUROC = 0.378; mean ABA to ST AUROC = 0.607) (Fig 3D–3F). Altering our analysis approach by averaging the 100 CFS feature sets, we again see a similar pattern in cross-dataset performance (ST to ABA difference in mean AUROC = 0.062; ABA to ST difference in mean AUROC = 0.381) (S7C–S7E Fig). These CFS results indicate that the observed high performance of classification within the ABA and lack of generalization to the ST is not driven by our choice of model. In summary, across both techniques, marker genes can be found to classify pairwise leaf brain areas from each other, but they often do not generalize to the opposite dataset.

**The sagittal subset of the ABA is the most distinct.** With only 2 datasets, it is impossible to distinguish whether the above lack of bidirectionality in cross-dataset learning is driven by (1) the ST being more generalizable or (2) a lack of information in ST that is critical to the high classification performance within ABA. To begin to address this, we took advantage of the separability of the ABA dataset into 2 distinct datasets: coronal and sagittal. The Allen Institute

collected duplicates of many genes; roughly 4,000 genes were collected across both the coronal and sagittal planes of slicing. With these 2 datasets alongside the ST, we further filtered for 3,737 overlapping genes across the same 445 leaf brain areas (see Methods) and computed all pairwise combinations of cross-dataset learning. Notably, using LASSO (lambda = 0.1), training on ST outperforms either plane of ABA in cross-dataset predictions: (1) ST to ABA coronal (mean AUROC = 0.888) performs better than ABA sagittal to ABA coronal (mean AUROC = 0.775); and (2) ST to ABA sagittal (mean AUROC = 0.671) performs better than ABA coronal to ABA sagittal (mean AUROC = 0.663) (Fig 3G). Further, the performance of models trained in ABA coronal to ABA sagittal (mean AUROC = 0.663) and ST to ABA sagittal (mean AUROC = 0.671) is lower than that of ABA coronal and ST to each other (ST to ABA coronal mean AUROC = 0.888; ABA coronal to ST mean AUROC = 0.796) (Fig 3G). This shows that the ABA coronal and ST are able to generalize to each other better than to the ABA sagittal. Across parametrizations of our model, the sagittal subset of the ABA continues to be the most distinct of the 3 datasets with the least generalizability (S8A and S8B Fig). To evaluate whether our selection of lambda had a significant impact on these findings, we looked at a subset of brain areas with larger sample sizes (minimum of 100) to allow dynamic LASSO hyperparameter fitting and compared it with a fixed hyperparameter (lambda = 0.1) in the same brain areas. This showed that performance was very similar between the two (S8C and S8D Fig) (see Methods).

The relative distinctness of the ABA sagittal dataset could be driven by its sparsity—consisting of zeros for more than half of the dataset (53.9%) compared to only 7.5% zeros in the coronal subset. LASSO is able to find a robust set of marker genes within the ABA sagittal that does not reflect the best possible set of genes in the less sparse ABA coronal and ST. While the coronal subset of the ABA was curated for genes showing spatial patterning [17], the subset of the sagittal genes in this analysis contains only those also present in the coronal set. So, the lack of generalizability of the sagittal subset is particularly suggestive of technical experimental or downstream processing issues rather than the absence of spatial patterning in the genes themselves.

## Distance in semantic space, but not physical space, provides a potential explanation for cross-dataset performance

Since the ARA brain areas are organized into a hierarchical tree-like structure based on biology [17], we hypothesized that the semantic distance of any 2 pairwise brain areas in this tree could provide an explanation for the cross-dataset performance of classifying samples from the same 2 areas. To investigate this, we used the path length of traversing this tree to get from one brain area to the second area as the measure of distance in the tree (see Methods). For the performance of classifying brain areas in both the ST and ABA when trained in the opposite dataset (LASSO, lambda = 0.1), we see an increase in performance (ST to ABA mean AUROC = 0.690 increases to mean AUROC = 0.912; ABA to ST mean AUROC = 0.655 increases to mean AUROC = 0.756) as the semantic distance increases from the minimum value of 2 to the maximum of 15 (Fig 4A and 4B). As expected, the corresponding increase in performance and semantic distance holds across parameterizations of our linear model (S9A–S9D Fig). A high AUROC here indicates that the 2 brain areas are transcriptionally distinct, while an AUROC near 0.5 indicates that they are similar. So, this result implies that distance in semantic space defined by the ARA reflects distance in expression space. This suggests that differences in classification performance are likelier to reflect real differences in gene expression between brain areas and not just large-scale gradients of expression present in the brain [33].

To further understand the relationship between performance and semantic distance, we next investigated pairs of brain areas with extreme AUROCs at the minimum and maximum

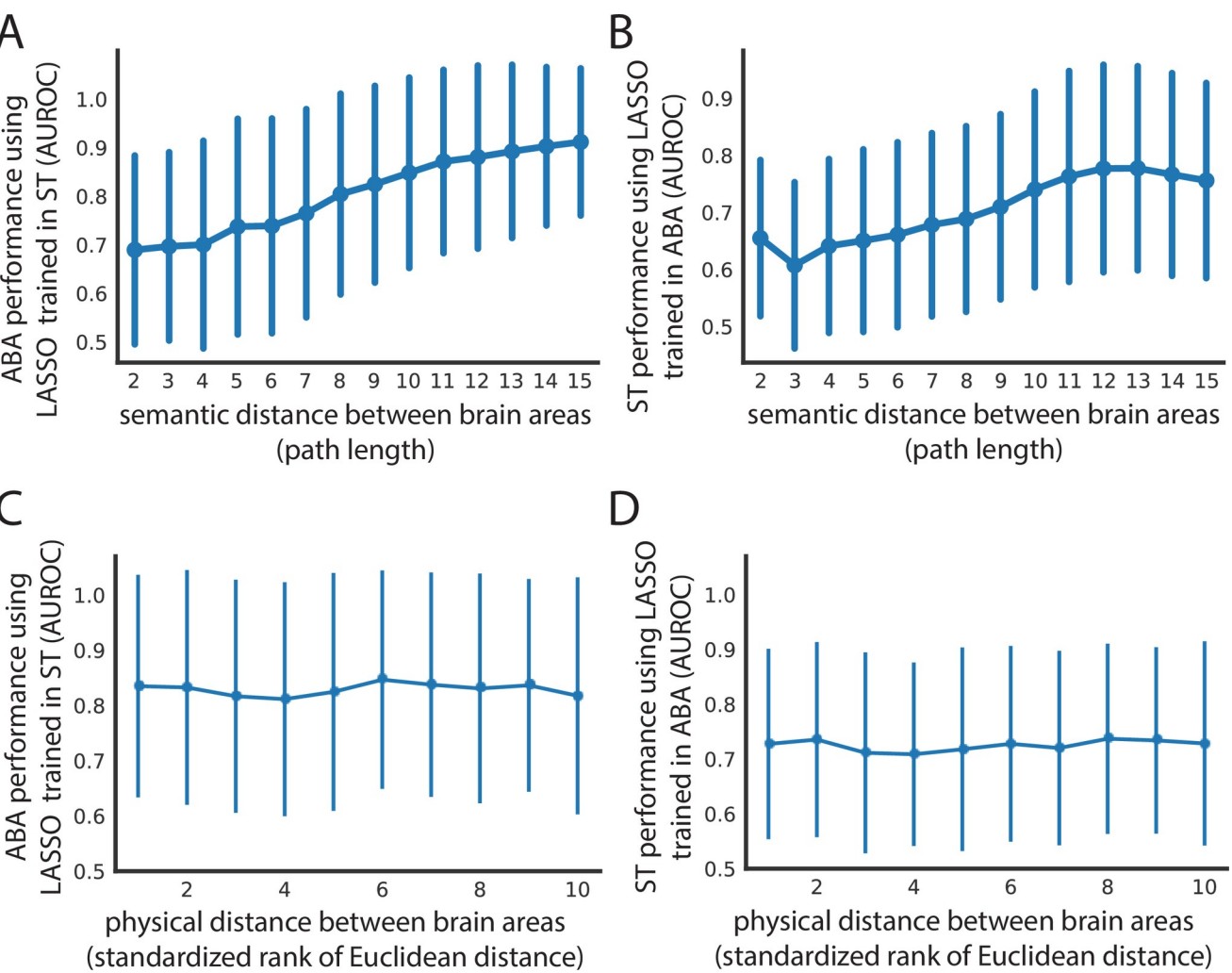

**Fig 4. Spatial expression patterns reflect distance in semantic space, but not physical distance in the brain.** Cross-dataset AUROCs (x-axis) of classifying all leaf brain areas from all other leaf brain areas for (A) ABA using LASSO (lambda = 0.1) trained in ST and (B) ST using LASSO (lambda = 0.1) trained in ABA as a function of path length (x-axis) in the ARA naming hierarchy between the 2 brain areas being classified. The same AUROCs (y-axis) from (A) and (B) shown in (C) and (D), respectively, as a function of minimum Euclidean distance between the 2 brain areas in the ARA (x-axis). Euclidean distance on the x-axis is binned into deciles for visualization. All 4 plots show mean AUROCs (points) with standard deviation (vertical bars). ABA, Allen Brain Atlas; ARA, Allen Reference Atlas; AUROC, area under the receiver operating curve; LASSO, least absolute shrinkage and selection operator; ST, spatial transcriptomics.

semantic distances. We were especially interested in this, given the distribution of AUROCs for each distance (Fig 4A and 4B). Similarly, at the smallest semantic distance of 2, in both ABA and ST trained in the opposite dataset, there is a spread in classification performance (S2 and S3 Tables). In both datasets, these brain area pairs involve different cortical layers of the same cortical area. The ARA hierarchy is organized such that within one cortical area, all the layers will have a semantic distance of 2 between each other. So, a pair of brain areas with a high AUROC and semantic distance of 2 often involves 2 nonneighboring layers of a cortical area (i.e., primary auditory cortex layer 6b and layer 4 in ST trained in ABA) (S3 Table). This trend is in line with our expectation as cortical layers are known to have distinct expression profiles driven in part by distinct cell types [34–37]. Alternatively, a pair of brain areas with an AUROC near 0.5 and a semantic distance of 2 can involve 2 neighboring layers of a cortical area (i.e., primary visual area layer 6a and layer 6b in ST trained in ABA) (S3 Table). This, too,

is not surprising because, despite distinctness in cortical layer expression, we expect some overlap between physically neighboring areas in terms of expression profiles due to errors introduced in sampling and in registration to the reference atlas. Together, these examples illustrate one way in which semantic distance is not synonymous to physical distance.

Since semantic distance does not perfectly capture the actual distance between brain areas, we next looked at classification performance as a function of physical distance directly. Specifically, we asked: Is performance in classifying pairwise leaf brain areas cross-dataset being driven by physical proximity/distance alone? Cross-dataset performance was examined with respect to the minimum Euclidean distance between the 2 brain areas in the ARA (see Methods). There is no trend between physical distance and AUROCs from either cross-dataset assessment using LASSO models trained in the opposite dataset (ABA to ST Pearson's r = −0.026; ST to ABA Pearson's r = 0.056) with the mean performance remaining similar at the minimum (ABA to ST mean AUROC = 0.651; ST to ABA mean AUROC = 0.702) and maximum distance (ABA to ST mean AUROC = 0.697, change in AUROC = +0.046; ST to ABA mean AUROC = 0.690, change in AUROC = −0.012) (Fig 4C and 4D) (see Methods). Across model parameterizations, there is similarly no relationship between distance and performance (S9E–S9H Fig). This result, alongside the positive relationship seen between performance and semantic distance, shows that spatial patterning of gene expression captures canonical brain area labels and is not merely composed of differences in large-scale gradients.

## Finding a uniquely identifying gene expression profile for individual brain areas

**Within one dataset, a gene expression profile can uniquely identify one brain area, but it does not generalize to the opposite dataset.** Thus far, we have focused on the classification of leaf brain areas from other leaf brain areas. However, this does not determine if we can uniquely identify a given brain area from the whole brain using gene expression. If possible, this could yield a set of marker genes to identify brain areas at their smallest parcellation for future neuroscience experiments. To tackle this, we trained linear models for one leaf brain area against the rest of the brain (one versus all) and tested that same model's performance in classifying the same leaf brain area against all others (one versus one across all leaf brain areas) (Fig 5A). Unfortunately, for most leaf brain areas, LASSO fails to fit a model with very light regularization (lambda = 0.01) to classify it against the rest of the brain in both the ST (mean train AUROC = 0.554) and the ABA (mean train AUROC = 0.593) (S10A–S10D Fig). The few leaf brain areas that are able to be classified from the rest of the brain using LASSO have a nearly identical performance in the one versus all case as in testing against all other leaf brain areas (Fig 5B, S10A and S10B Fig). At a higher regularization weight (lambda = 0.05), most one versus all models fail to be trained (ST mean train AUROC = 0.501; ABA mean train AUROC = 0.502) (Fig 5B, S10E–S10H Fig). Failing to find potential marker genes using this approach with regularized LASSO, we turned to unregularized linear regression (i.e., lambda = 0), with the hope to minimally find an identifying expression profile. Using linear regression, performance of models fit in the one versus all case correlates nearly perfectly with the average performance of the same model in one versus one. This nearly identical performance is true in both the ST (mean distance from identity line = 0.005) and ABA datasets (mean distance from identity line = 0.001) (Fig 5B–5D) (see Methods). This result demonstrates that within a dataset, we can find an identifying gene expression profile of a brain area that uniquely identifies it.

Since we could robustly identify a gene expression profile to identify a brain area within one dataset, we next asked if these profiles can generalize to the opposite dataset. Using the

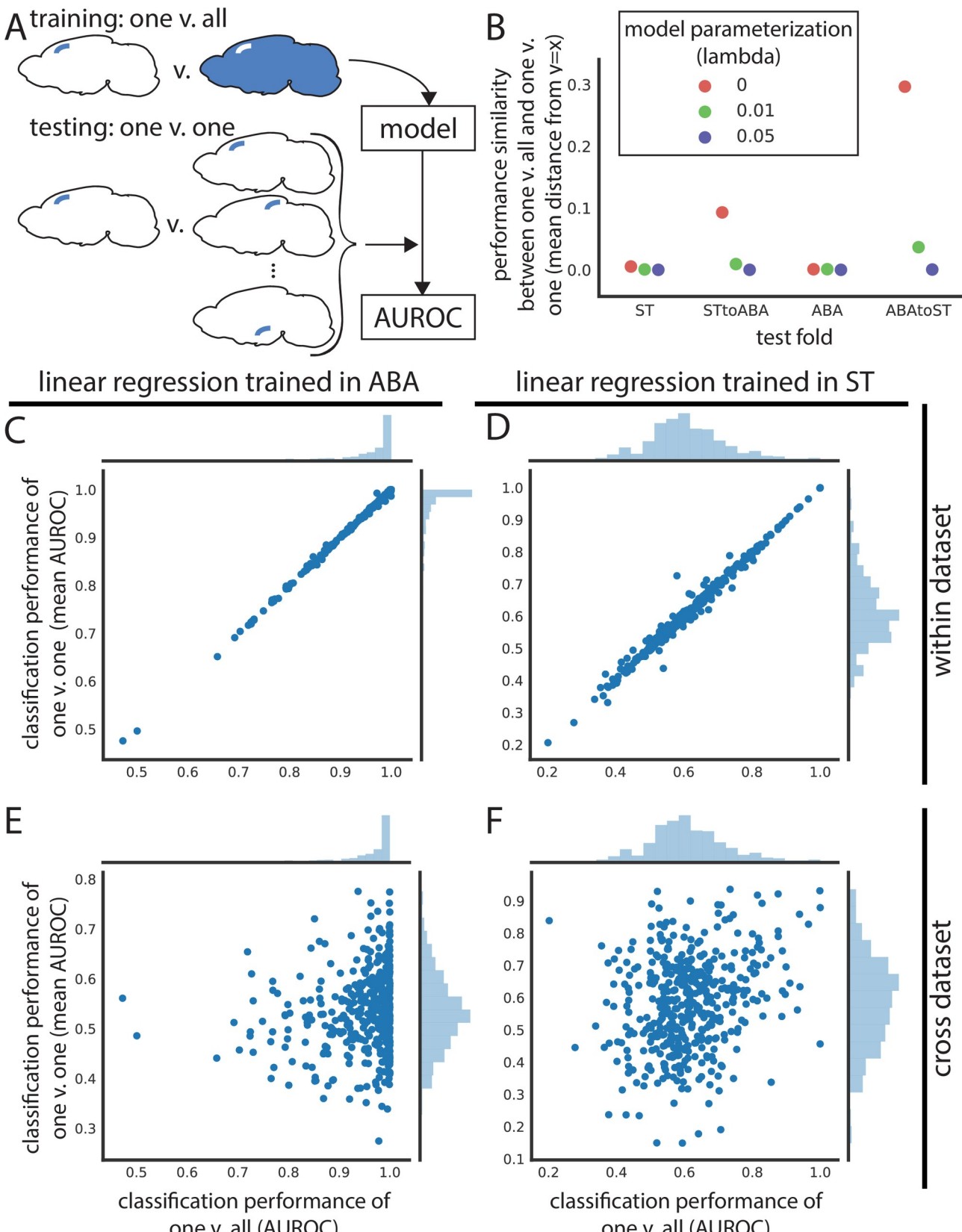

**Fig 5. Leaf brain area expression profiles are identifiable within dataset but do not generalize cross-dataset.** (A) Schematic depicting training and testing schema for panels in this figure. Models are trained to classify one leaf brain area against the rest of the brain (one vs. all) and then used to test classification of that brain area against all other leaf brain areas (one vs. one) within and across dataset. (B) Performance similarity between one vs. all and one vs. one reported as the mean absolute value of distance from identity line for scatter plots of testing in one vs. all against one vs. one. Performance similarity shown for within ST, ST to ABA, within ABA, and ABA to ST across linear regression (red), LASSO (lambda = 0.01) (green), and LASSO (lambda = 0.05) (violet). Linear regression one vs. all test set performance (x-axis) vs. average one vs. one performance of the same model (y-axis) in (C) ABA and (D) ST. (E) Assessment of the same ABA one vs. all linear regression model (x-axis) in one vs. one classification in the ST dataset (y-axis). (F) Same as (E), but one vs. all linear regression trained in ST (x-axis) and one vs. one classification of these models in ABA (y-axis). ABA, Allen Brain Atlas; AUROC, area under the receiver operating curve; LASSO, least absolute shrinkage and selection operator; ST, spatial transcriptomics.

same models trained in one versus all in either the ST or ABA, we classified the same brain area against all other brain areas (one versus one) in the second dataset. The one versus all trained linear models (lambda = 0) do not generalize cross-dataset for either ABA to ST (mean distance from identity line = 0.296) or the reverse (ST to ABA mean distance from identity line = 0.093) (Fig 5B, 5E, and 5F). This lack of cross-dataset performance similarly holds for other parameterizations (Fig 5B, S10C, S10D, S10G, and S10H Fig). The identifying gene expression profile of a leaf brain area is not generalizable to a new dataset that is not used in defining that profile. So, while we can uniquely identify a brain area using gene expression within one dataset, that identification profile does not extend to the second dataset.

## Conclusions

Across disciplines, benchmarking studies have helped to advance their respective fields and set standards for future research [20,38–40]. Neuroscience is no exception. Given the complexity of the brain, however, the addition of multimodal information is especially desirable. Studying the brain from a variety of perspectives, across data modalities, technology platforms, and experiments, can give a complete, composite understanding of its biology [7]. Coupled with the stereotyped substructure and transcriptional heterogeneity, the adult mammalian brain is the ideal model system to assess new spatial gene expression technologies. With this in mind, we linked 2 modalities to ask: Can we capture canonical, anatomically defined brain areas from the ARA using spatial gene expression alone? And, how well does this replicate across 2 transcriptomic datasets collected using different platforms?

Principally, we showed that ARA brain labels are classifiable using only gene expression but highlighted a lack of generalizability across spatial transcriptomic datasets. Within datasets, we are able to distinguish brain areas from each other with high performance. We were further able to uniquely identify a brain area within dataset; training on one brain area against the entire rest of the brain generalizes to testing on that same brain area against all other leaf brain areas. Notably, within-dataset performance was on average higher in the ABA than in the ST, which led to a lack of cross-dataset generalizability when training in the ABA and testing in ST; this phenomenon was not present in reverse. However, in both cases, there is an observed trend linking an increase in mean cross-dataset performance with increased semantic distance in the ARA brain area label organization. There was no link in performance when compared to physical distance in the brain, suggesting that ARA labels are meaningful in expression space and we are not simply detecting spatial differences in gene expression.

It is important to point out that our benchmarking study at its core only involves 2 independent datasets, although both are extraordinary in scope. Limited to 2 datasets, it is impossible to tell whether one dataset or the other is closer to representing the ground truth. Further, the ABA and ST datasets use 2 fundamentally different techniques: The ABA data report average pixel intensity from ISH, and the ST approach is an RNA capture technique followed by sequencing that reports read counts. In addition, the spatial resolution varies between the 2 datasets. Each sample in the ST is further apart within a plane due to a 200-μm center-to-

center distance for probe spots [1] in comparison to samples in the ABA ISH with 200 μm$^3$ voxels that tile adjacently [17]. Conversely, the ABA ISH has a lower Z resolution or larger gap between slices (200 μm) when compared to the ST (median slicing period of 100 μm) [21]. Further, after the collection of the raw expression data, each of the 2 datasets also undergoes a unique registration step to the ARA. The ABA uses an iterative approach that involves registration of the 3D brain volume with interspersed smoothing steps [18], while the ST dataset is registered on a slice-by-slice basis to the nearest representative 2D ARA slice using anatomical landmarks [41]. Beyond registration, there are additional concerns about the stability of the ARA brain area labels since there are inconsistencies with other brain atlases and even across versions of the ARA [42–44]. Brain atlases are an imperfect formalism of brain substructure but are the best systematic representation to test spatial gene expression by biological areas.

Past these technical differences, it is also possible that the lack of strong cross-dataset generalization could represent true biological brain-to-brain variability of individual mice. Slight changes in cellular composition between individuals near borders of brain areas could be responsible for the differences between these areas. Zooming out, the results of this manuscript have many potential implications for neuroscience. First, we observed that brain regions are comprehensively better defined by a combination of many genes as opposed to individual markers. Traditionally, however, the description and/or subsequent experimental identification of brain areas by gene expression, often via specific populations of neurons, usually depended on 1 or 2 marker genes [45,46]. The choice to use single marker genes is usually one of practicality, but as spatial experimental and computational analysis techniques continue to improve, the possibility of using better resolved brain areas defined by a multigene expression profile is within reach [21]. This in turn could inform downstream experimental identification of brain areas. Secondly, our results show that quantitative exploration of the existence of brain regions is timely, just as single-cell data have made quantitative exploration of cell type definitions timely. The definition of brain regions is necessary to study the brain, but existing parcellations are almost certainly not sufficient; it is easy to overfit to rigidly defined areas, glossing over individual differences. This highlights the need for continued development of approaches for validating and integrating spatial data from multiple sources. The ST data are a technology that is timely to integrate with single-cell data and valuable to validate, refine, and discover an iterative neuroanatomy that grows with new data types and sources [47]. Finally, an iterative, multimodal definition of brain areas could aid in the research of cross-species comparisons and disease phenotypes, where the mapping of neuroanatomical landmarks is potentially complex.

As the types and prevalence of spatial gene expression approaches continue to increase [5], whole-brain spatial gene expression datasets will surely follow. By continuing to integrate these emerging datasets, we will be able to perform more robust meta-analyses, giving us a deeper understanding of both spatial gene expression with respect to ARA labels and the replicability of spatial technologies in general. An added benefit of the continued incorporation of additional datasets is that, at some point, differences in experimental platforms and registration approaches will only contribute to the robustness of any biological claims. We believe that continued meta-analysis of spatial gene expression in the adult mouse brain and other biological systems is an important route toward integration of distinct data types—location and expression—to form the beginnings of a robust, multimodal understanding of the mammalian brain and other systems.

## Methods

### Spatial transcriptomics (ST) data

ST is an array-based approach where a tissue section is placed on a chip containing poly-T RNA probes that the mRNA transcripts present in the tissue can hybridize to [1]. These probes

tile the chip in 100-μm diameter spots and contain barcodes specific to that spot so that RNA sequencing reads can be mapped back to their original grid location. Note that the probe spots are not perfectly adjacent to each other but have a center-to-center distance of 200 μm [1].

Here, we used a previously published spatial gene expression dataset containing 75 coronal slices from one hemisphere of the adult mouse brain across 3 animals [21]. The coronal slices were mapped to the Allen Mouse Brain Reference Atlas using a nonrigid transformation approach [41]. In total, this dataset contains 34,103 ST spots across 23,371 genes [21].

### Allen Brain Atlas (ABA) in situ hybridization data

The ABA adult mouse ISH dataset consists of a transcriptome-wide assay of expression in inbred WT mice using single-molecule ISH [17]. To assay the whole transcriptome, many WT mouse brains were sliced into 25-μm thick slices containing 8 interlayered sets for subsequent single-molecule hybridization or for staining to create the reference atlas. This results in a z resolution of 200 μm for each gene. These independent image series are subsequently reconstructed to 3 dimensions and registered to the reference brain atlas in interlayered steps [18]. There are 26,078 series, or experiments, across both coronal and sagittal planes with 19,942 unique genes represented. This suggests that a minimum of roughly 3,260 mice brains were used in this dataset, which does not include series that were unused or used for reference staining. These 3D registered reconstructions are then segmented to 200 $\mu m^3$ voxels with an associated brain area label. There are 159,326 voxels, with 62,529 mapping to the brain. Gene expression for each of the assayed genes was quantified in these voxels from the imaged data as energy values, which is defined as the sum of expression pixel intensity divided by the sum of all pixels.

The quantified ISH energy values dataset was downloaded from the ABA website (http://help.brain-map.org/display/mousebrain/API) through their API on March 12, 2019.

### Allen Institute reference brain ontology, leaf brain areas, and path length

The Allen Institute reference brain atlas has organized brain areas into a hierarchy described by a tree data structure. Leaf brain areas are defined here as brain areas that constitute leaves on the ontology tree, i.e., they have no children. Leaf brain areas represent the most fine-scale parcellation of the brain. Using leaf brain areas circumvents the fact that the depth of the tree representing the hierarchical naming structure of brain areas in the ARA is not uniform. Path length refers to the number of steps required to go from one brain area to another in this tree.

### Data filtering and train/test split

The ST data were preprocessed to remove ST spots mapping to ambiguous regions, fiber tracts, or ventricular systems and to remove genes that were expressed in less than 0.1% of samples. This left 30,780 ST spots, or samples, with 16,557 genes. For within-ST analyses, this dataset was further filtered to 461 leaf brain areas that each had a minimum of 5 spots. In all analyses, these spots are subsequently randomly split into train and test sets with a 50/50 split. The train/test split is random but stratified for brain areas so that each fold has roughly 50% of the samples belonging to each brain area. N-fold (here, 2) cross validation was used, and results are reported as a mean across folds.

Similarly, the ABA data were filtered for only voxels mapping to the reference brain and genes with expression in at least 0.1% of samples. This gives 26,008 series across 62,527 voxels, also split as described for ST into 50/50 train and test folds. There are 4,972 genes that are assayed more than once across independent experimental series. Except for the analyses

separating out the 2 planes of the ABA data (detailed below), genes duplicated across series were averaged for each voxel for a total of 19,934 unique genes. For within-ABA analyses, this dataset was further filtered to 560 leaf brain areas that each had a minimum of 5 voxels prior to the train/test split. As with ST, within-ABA training, and testing, n-fold (here, 2) cross validation was used, and results are reported as a mean across folds.

For cross-dataset learning, both datasets were further filtered for 445 leaf brain areas that were represented with a minimum of 5 samples in each dataset. Genes were also filtered for those present in both datasets resulting in 14,299 overlapping genes between the two. This filtered subset was used for cross-dataset test set classification and matched within-dataset test set comparisons. For analyses separating out the 2 ABA planes, a similar mapping process was used to determine overlaps between each of the planes and the ST data. This resulted in 3,737 overlapping genes across the same 445 leaf brain areas. Genes that were duplicated in the ABA dataset with independent imaging series within a plane were averaged.

## Area under the receiver operating characteristic (AUROC), clustering using AUROC, and precision

The AUROC is typically thought of as calculating the area under the curve of true positive rate as a function of false positive rate. Here, the area under the empirical ROC curve is calculated analytically since it is both computationally tractable and accurate for a given sample. It is given by

$$AUROC = \sum_i^N \frac{Ranks_i}{N_{Pos} * N_{Neg}} - \frac{N_{Pos} + 1}{2 * N_{Neg}}$$

where ranks are the ranks of each positive label sorted by feature, and $N_{Pos}$ and $N_{Neg}$ are the number of positive and negative labels, respectively. This formula is based on the relationship between the Mann–Whitney $U$ (MWU) statistic and AUROC [48–50]. An AUROC of 0.5 indicates that the task being evaluated is performing at chance, while an AUROC of 1 indicates perfect performance. For within-dataset analysis (Fig 2), any AUROCs of 0 were removed from downstream reporting of distributions and mean AUROCs. Note, this filtering does not alter the reported means to the third decimal place. For within-dataset analyses, AUROCs are reported as the mean across 2-fold cross validation.

Clustering by AUROC is done by converting AUROC to a similarity metric by subtracting 0.5 to center the AUROC values at 0.5 and taking the absolute value. The rationale is that if a classification task performs with an AUROC of 0.5, the 2 classes are so similar that they are not distinguishable so they should be grouped closely.

Here, we calculate precision for the median performing brain area pair given by AUROC for within-dataset analysis. We use a threshold that includes all instances of one class, here, all instances of one brain area. Precision is calculated as:

$$precision = \frac{true\ positives}{(true\ positives + false\ positives)}$$

Note that the AUROC of median performing brain area pairs are calculated from the averaged AUROCs across 2 folds, while the reported precision is the average precision of all median brain area pairs from each fold independently because the reported median AUROC (from the fold averaged AUROCs) does not match to actual brain area pairs in either fold.

## LASSO and penalty hyperparameter selection

Least absolute shrinkage and selection operator, or LASSO regression, uses an L1 penalty for fitting the linear regression model [25]. The cost function to minimize is given by:

$$cost\ function = \min_{\omega} \frac{1}{2n_{samples}} \|X\omega - y\|_2^2 + \lambda \|\omega\|_1$$

where X represents the matrix of feature values, y the target values, $\omega$ the coefficients, and $\lambda$ the constant value with which to weight the regularization. The notation $\|.\|_1$ represents the L1 norm. A small $\alpha$ gives little regularization ($\alpha = 0$ is equivalent to regular linear regression). An L1 penalty minimizes the absolute value of coefficients, which has an effect of pushing many coefficients toward zero. This is beneficial for highly correlated data to find an optimal set of features among correlated genes, or features, to use for prediction.

In this manuscript, LASSO models are fit using coordinate descent according to the scikit-learn library [51]. Hyperparameter selection for the penalty weight $\lambda$ is done through cross validation on a subset of brain areas that have sufficient sample size; we use a cutoff of having greater than 100 samples per brain area, which resulted in 65 areas in ST and 139 areas in ABA. With this subset, we use the StratifiedShuffleSplit function from the scikit-learn library to create 3 folds with a test size of 20% within the 50% train set for each dataset [51]. These folds can overlap with each other but are random and stratified by label. We next use these folds in the GridSearchCV function of scikit-learn to perform hyperparameter selection over $\lambda$ values of 0.01, 0.05, 0.1, 0.2, 0.5, and 0.9. Note, when returning the best-performing hyperparameter or classification result as an AUROC, GridSearchCV returns the first of ties, which can be misleading with tied performance across many hyperparameters (as is often the case here). In both ST and ABA, most pairwise brain area LASSO models perform best with the smallest given $\lambda$ of 0.01 (S11A and S11B Fig). Since most brain areas lack the sample size to dynamically fit alpha, we chose a fixed $\lambda$ value for all brain areas in our brain-wide analyses. Although there is not a clear trend, and keeping the ties or near ties in performance in mind, we use a $\lambda$ of 0.1 for most of our analyses as larger lambda values tend to only show up for smaller brain areas. The hyperparameter $\lambda$ used for each analysis is noted throughout the main text. We further perform hyperparameter selection in the cross-dataset case when the planes of ABA are separated out. Using 63 brain areas with greater than 100 samples across all 3 "datasets," we find that mean AUROCs across pairwise cross-dataset classification was comparable between the dynamically fitted $\lambda$ and the fixed $\lambda = 0.1$ across brain area pairs (S8C and S8D Fig). For additional details on parameterization, see code scripts (repository availability below).

## Linear regression

Linear regression is implemented using scikit-learn with default parameters [51]. Normally, when there are more features than samples, linear regression is underdetermined. In the scikit-learn library, however, instead of returning linear regression as unsolvable, it returns the minimum Euclidean norm. (This is different from Ridge Regression where the L2 norm is incorporated in the cost function.)

## K-nearest neighbors (k-NN) algorithm

For applications of k-NN, we used the scikit-learn implementation with default hyperparameters: k = 5, weights = "uniform," algorithm = "auto" [51]. Similar to LASSO, 50% of the data was used as the train set, calculating performance of classification on the other 50% held-out test set. As an output, k-NN gives a 1D vector with the length equal to the number of samples

in the test set. This vector contains a predicted brain area label for each test set sample based on the most highly represented class among each test set sample's k closest neighbors in expression space. To compare this classification result to our other classification approaches, we separated out the 1D vector into a 2D binary matrix with a column for each brain area and rows of the same length as the 1D vector representing samples. Each time a sample is predicted as being a particular brain area, the corresponding row and column are marked with a 1. This matrix is then used to calculate an AUROC for predicting each brain area, or column. The mean AUROC of these brain areas is reported in the manuscript.

## Batch correction using pyComBat

For batch correction, we use pyComBat, a recent python-based implementation of ComBat [52–54]. Prior to batch correcting, we normalize each dataset independently using z-scoring. We then run pyComBat on the 2 datasets combined treating each dataset as a batch. We do not include covariates. Corrected data are then parsed into the 2 datasets from the combined matrix for subsequent cross-dataset LASSO analysis.

## Assessing dimensionality of data using principal component analysis (PCA)

PCA, as implemented in scikit-learn [51], was used to determine the dimensionality of both datasets. PCA was applied to the genes, or features, for each leaf brain area separately in the ABA and ST datasets. The total number of components to use for dimensionality reduction was set to be equal to the number of samples in each area. ABA areas were down-sampled to have the same number of samples as the corresponding brain area in ST. There are 77 brain areas that exceptionally have fewer samples in ABA than ST, so when down-sampling ABA for these 77 areas, the original sample size was used. Dimensionality of the brain areas was then accessed as the number of PCs needed to explain at least 80% of the variance.

## Differential expression and correlation-based feature selection (CFS)

Differential expression (DE) in genes is assayed using MWU. Resulting *p*-values are not corrected for multiple hypothesis testing since *p*-values are only used to threshold for very extreme DE genes across brain area comparisons. The uncorrected *p*-values themselves are not reported as a measure for significant DE.

CFS is a feature selection technique that explicitly picks uncorrelated features [32]. Here, a greedy approach to CFS was implemented. The algorithm first chooses a random seed or gene within the top 500 differentially expressed genes. The next gene is then chosen as the lowest correlated gene to the first one and kept if the set AUROC improves. Subsequent genes are chosen as the least correlated on average to the genes already in the feature set. The algorithm stops once the AUROC is no longer improving. The final set of genes chosen using CFS are then aggregated by equally averaging the values of all chosen genes for each sample. Here, in particular, these feature sets fell in the range of 1 to 29 genes with a median of 2 genes in the ABA and the range of 1 to 47 genes with a median of 4 genes in the ST (S7A and S7B Fig). For more details on exact implementation, see code scripts (repository available below).

For the cross-dataset analysis, when unspecified, 100 feature sets were chosen using this approach, and the single best-performing feature set was then evaluated in both the within-dataset test set and the cross-dataset test set. When indicated accordingly, the 100 CFS feature sets were averaged instead of reporting the performance of the best set alone.

## Euclidean distance between 2 brain areas

In addition to brain area labels, the ABA dataset contains x, y, z coordinates for each voxel in the ARA space. So, physical distance between 2 brain areas is calculated as the Euclidean distance between the 2 closest voxels where each voxel belongs to one or the other brain area. Due to the symmetry of brain hemispheres, distance was only calculated in one hemisphere by filtering for voxels with a z-coordinate less than 30. This z-coordinate was visually determined to be the midline of the brain based on 3D visualization of the voxel coordinates. Euclidean distances between brain areas calculated in this manner were used for both the ST and ABA datasets since both are registered to the ARA.

## Mean distance from identity line

To assess the replicability of models trained in one brain area versus the rest of the brain (one versus all) in classifying that same brain area against all the others (one versus one), the mean absolute Euclidean distance of a scatter plot of those 2 values from the identity line was calculated. This was done to assess how similar the values in the one versus all case are to the one versus one case for each pair of brain areas. Correlation was found to be lacking because it could yield high correlations when the one versus all and one versus one values were quite different for a given point.

## Code

All code used for the analyses described in this manuscript was written in Python 3.7 with supporting packages: jupyterlab 1.0.9, h5py 2.9.0, numpy 1.16.4, scipy 1.3.1, pandas 0.25.0, scikit-learn 0.21.2, matplotlib 3.1.0, and seaborn 0.9.0. All Jupyter notebooks and scripts are available on GitHub at www.github.com/shainalu/spatial_rep.

## Supporting information

**S1 Fig. Additional visualization and verification of within-dataset LASSO results with a new random train/test split.** Histogram of classification performance of LASSO (lambda = 0.1) in (A) ABA test fold and (B) ST. (A) and (B) represent the upper triangular of Fig 2A and Fig 2B, respectively. Black dashed vertical line represents the mean. Heat map of AUROC for classifying leaf brain areas from all other leaf brain areas in (C) ABA and (D) ST using LASSO (lambda = 0.1) using a different random train/test split with a seed = 9 relative to Fig 2A and 2B. Dendrograms on the far left side represent clustering of leaf brain areas based on the inverse of AUROC; areas with an AUROC near 0.5 get clustered together, while areas with an AUROC near 1 are further apart. Color bar on the left represents the major brain structure that the leaf brain area is grouped under. These areas include CTX, MB, CB, CNU, HB, and IB. Histogram of classification performance of LASSO (lambda = 0.1) in (E) ABA test fold and (F) ST. (E) and (F) represent the upper triangular of (C) and (D), respectively. Black dashed vertical line represents the mean. ABA, Allen Brain Atlas; AUROC, area under the receiver operating curve; CB, cerebellum; CNU, striatum and pallidum; CTX, cortex; HB, hindbrain; IB, thalamus and hypothalamus; LASSO, least absolute shrinkage and selection operator; MB, midbrain; ST, spatial transcriptomics.
(TIF)

**S2 Fig. LASSO performance with permuted labels falls to chance and k-NN performance.** Upper triangular (A) histogram and (C) heat map of AUROC for classifying leaf brain areas from all other leaf brain areas in ABA using LASSO (lambda = 0.1) when brain area labels are randomly permuted. (B) and (D) same as (A) and (C), respectively, but for ST with labels

permuted. For heat maps (C, D), dendrograms on the far left side represent clustering of leaf brain areas based on the inverse of AUROC; areas with an AUROC near 0.5 get clustered together, while areas with an AUROC near 1 are further apart. Color bar on the left represents the major brain structure that the leaf brain area is grouped under. These areas include CTX, MB, CB, CNU, HB, and IB. Distribution of performance (AUROC) of classifying each brain area using k-NN, a multiclass classifier, with default parameters (k = 5) for (E) ABA and (F) ST. Mean AUROC is shown in the upper left of each plot. ABA, Allen Brain Atlas; AUROC, area under the receiver operating curve; CB, cerebellum; CNU, striatum and pallidum; CTX, cortex; HB, hindbrain; IB, thalamus and hypothalamus; k-NN, k-nearest neighbors; LASSO, least absolute shrinkage and selection operator; MB, midbrain; ST, spatial transcriptomics. (TIF)

**S3 Fig. Classification using single genes, relative expression across datasets, and PCA.** Distribution of classifying (A) CA2 and (B) arcuate hypothalamic nucleus against the rest of the brain using single genes. Distributions of all single genes shown for classification in ABA (blue) and ST (orange). Dashed lines represent the marker gene (A) Amigo2 or (B) Pomc for classification in ABA (blue) or ST (orange). (C) Relative expression between the ST and ABA datasets as a density plot. Expression is plotted as the ranked mean for each gene across all samples. (D) Cumulative explained variance curves for PCA in ST (orange) and ABA (blue). Each curve represents one leaf brain area. The total number of principal components per brain area is equal to the number of samples in that area. ABA areas that had more samples than ST are randomly down-sampled accordingly. For both datasets, the Caudoputamen, the largest region, is removed to allow visualization; full figure shown in inset plot. (E) Cumulative explained variance curves for 200 PCs in the whole ST (orange) and whole ABA (blue) datasets. ABA, Allen Brain Atlas; AUROC, area under the receiver operating curve; PCA, principal component analysis; ST, spatial transcriptomics. (TIF)

**S4 Fig. Sample correlation within brain areas and relationship between size and classification performance.** (A) Distribution of sample correlation within each of the leaf brain areas for ABA (blue) and ST (orange). Vertical dashed line represents the mean for the correspondingly colored distribution. Test set AUROC in (B) ST and (C) ABA as a function of the number of samples per brain area. The minimum of the 2 brain areas involved in classification is shown. ABA, Allen Brain Atlas; AUROC, area under the receiver operating curve; ISH, in situ hybridization; ST, spatial transcriptomics. (TIF)

**S5 Fig. Additional verification of cross-dataset LASSO results with a new random train/ test split.** (A) Models trained with overlapping genes and brain areas between ST and ABA datasets are evaluated within dataset on the test fold and across dataset on the entire opposite dataset as illustrated in Fig 1C. Summary diagrams showing mean AUROC for within-dataset test set performance (purple arrow) and cross-dataset performance with models trained in the opposite dataset (light blue arrow) for (A) LASSO (lambda = 0.1) with a new random train/test split (seed = 9) relative to Fig 3A–3C. Distributions of AUROCs for within- (purple) and cross-dataset (light blue) performance for (B) LASSO (lambda = 0.1) trained in ST with new train/test split and (C) LASSO (lambda = 0.1) trained in ABA with new train/test split. In both plots, dashed vertical lines represent the mean of the corresponding colored distribution. ABA, Allen Brain Atlas; AUROC, area under the receiver operating curve; LASSO, least absolute shrinkage and selection operator; ST, spatial transcriptomics. (TIF)

**S6 Fig. Visualization of the ABA and ST datasets together in low-dimensional space.** Plots showing ABA (blue) and ST (orange) datasets visualized together in low-dimensional space after dimensionality reduction with PCA. (A–C) show ST plotted on top of ABA, while (D–F) show the same PCs with ABA plotted on top of ST. Plots show (A, D) PC1 vs. PC2, (B, E) PC2 vs. PC3, and (C, F) PC1 vs. PC3. ABA, Allen Brain Atlas; PCA, principal component analysis; ST, spatial transcriptomics.
(TIF)

**S7 Fig. Feature set sizes for CFS and cross-dataset results for CFS with averaging across feature sets.** Distribution of CFS feature set sizes for (A) ABA and (B) ST. Models trained with overlapping genes and brain areas between ST and ABA datasets are evaluated within dataset on the test fold and across dataset on the entire opposite dataset as illustrated in Fig 1C. (C) Summary diagrams showing mean AUROC for within-dataset test set performance (purple arrow) and cross-dataset performance with models trained in the opposite dataset (light blue arrow) using the average of 100 feature sets chosen with CFS. Distributions of AUROCs for within- (purple) and cross-dataset (light blue) performance for 100 averaged CFS picked gene sets trained (D) in ST and (E) in ABA. In both plots, dashed vertical lines represent the mean of the correspondingly colored distribution. ABA, Allen Brain Atlas; AUROC, area under the receiver operating curve; CFS, correlation-based feature selection; ST, spatial transcriptomics.
(TIF)

**S8 Fig. Summary plots for cross-dataset analysis of ST, ABA coronal, and ABA sagittal with various parameterizations.** Separating out the 2 planes of slicing in the ABA and treating them alongside the ST dataset as 3 different datasets for cross-dataset learning. Summary diagram showing mean AUROCs using (A) linear regression and (B) LASSO (lambda = 0.05). (C, D) Same cross-dataset analysis as (A, B) but looking only at brain areas with at least 100 samples for (C) fixed lambda = 0.1 and (D) dynamically fitted lambda for each brain area pair. In all 4 summary diagrams, cross-dataset arrows originate from the dataset that the model is trained in and point to the dataset that those models are tested in. ABA, Allen Brain Atlas; AUROC, area under the receiver operating curve; LASSO, least absolute shrinkage and selection operator; ST, spatial transcriptomics.
(TIF)

**S9 Fig. Comparison of path length and Euclidean distance to LASSO performance for various parameterizations of LASSO.** Cross-dataset AUROCs (x-axis) of classifying all leaf brain areas from all other leaf brain areas for (A) ABA using LASSO (lambda = 0.05) trained in ST, (B) ABA using linear regression trained in ST, (C) ST using LASSO (lambda = 0.05) trained in ABA, and (D) ST using linear regression trained in ABA as a function of path length (x-axis) in the ARA naming hierarchy between the 2 brain areas being classified. The same AUROCs (y-axis) from (A–D) shown in (E–H), respectively, as a function of minimum Euclidean distance between the 2 brain areas in the ARA (x-axis). Euclidean distance on the x-axis is binned into deciles for visualization. All plots show mean AUROCs (points) with standard deviation (vertical bars). ABA, Allen Brain Atlas; ARA, Allen Reference Atlas; AUROC, area under the receiver operating curve; LASSO, least absolute shrinkage and selection operator; ST, spatial transcriptomics.
(TIF)

**S10 Fig. One vs. all and one vs. one analysis across various parameterizations.** LASSO (lambda = 0.01) one vs. all test set performance (x-axis) vs. average one vs. one performance (y-axis) of the same dataset (A) in ABA and (B) in ST. (C, D) Same as (A and B), but one vs. one performance (y-axis) is accessed in the opposite dataset for (C) train one vs. all in ABA

and test one vs. one in ST and (D) train one vs. all in ST and test one vs. one in ABA. (E–H) Same as (A–D), respectively, but using LASSO (lambda = 0.05). ABA, Allen Brain Atlas; AUROC, area under the receiver operating curve; LASSO, least absolute shrinkage and selection operator; ST, spatial transcriptomics.
(TIF)

**S11 Fig. Relationship between sample size and LASSO hyperparameter choice.** Plots showing the best lambda for LASSO when dynamically fit across various possible values as a function of brain area sample size in (A) ABA and (B) ST. ABA, Allen Brain Atlas; LASSO, least absolute shrinkage and selection operator; ST, spatial transcriptomics.
(TIF)

**S1 Table. Cross-dataset classification performance (mean AUROC) for batch-corrected vs. not-batch corrected ABA and ST datasets.** ABA data are randomly down-sampled here to have the same sample size as ST. Note that the not-batch corrected case is not down-sampled, but it is filtered for the brain areas that are included in the batch corrected mean AUROCs after down-sampling. ABA, Allen Brain Atlas; AUROC, area under the receiver operating curve; ST, spatial transcriptomics.
(PDF)

**S2 Table. Examples of brain area pairs from LASSO (lambda = 0.1) trained in ST and tested in ABA with minimum path lengths with low and high AUROCs.** ABA, Allen Brain Atlas; AUROC, area under the receiver operating curve; LASSO, least absolute shrinkage and selection operator; ST, spatial transcriptomics.
(PDF)

**S3 Table. Examples of brain area pairs from LASSO (lambda = 0.1) trained in ABA and tested in ST with minimum path lengths with low and high AUROCs.** ABA, Allen Brain Atlas; AUROC, area under the receiver operating curve; LASSO, least absolute shrinkage and selection operator; ST, spatial transcriptomics.
(PDF)

## Acknowledgments

The authors would like to thank Leon French for providing insight on systematically accessing the Allen Institute data, Manthan Shah for executing this, and Nathan Fox for streamlining datasets used. The authors would also like to thank Jose Fernandez Navarro for mapping the RNA-seq data of the Spatial Transcriptomics dataset. Finally, the authors would like to thank Sara Ballouz, Xiaoyin Chen, Megan Crow, Aki Funamizu, Benjamin Harris, Longwen Huang, Risa Kawaguchi, Elyse Schetty, Colin Stoneking, Jessica Tollkuhn, and Alex Vaughan for useful input and discussion.

## Author Contributions

**Conceptualization:** Anthony Zador, Jesse Gillis.

**Data curation:** Shaina Lu, Cantin Ortiz, Konstantinos Meletis.

**Formal analysis:** Shaina Lu, Daniel Fürth, Stephan Fischer, Konstantinos Meletis.

**Funding acquisition:** Anthony Zador, Jesse Gillis.

**Methodology:** Shaina Lu, Stephan Fischer, Konstantinos Meletis.

**Resources:** Anthony Zador, Jesse Gillis.

**Supervision:** Konstantinos Meletis, Anthony Zador, Jesse Gillis.

**Visualization:** Shaina Lu.

**Writing – original draft:** Shaina Lu.

**Writing – review & editing:** Shaina Lu, Stephan Fischer, Anthony Zador, Jesse Gillis.

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
