## [Editor Report · Decision Letter 0]

28 Jan 2021

Dear Dr Gillis, 

Thank you for submitting your manuscript entitled "Assessing the replicability of spatial gene expression using atlas data from the adult mouse brain" for consideration as a Research Article by PLOS Biology.

Your manuscript has now been evaluated by the PLOS Biology editorial staff, as well as by an academic editor with relevant expertise, and I am writing to let you know that we would like to send your submission out for external peer review.

Please re-submit your manuscript within two working days, i.e. by Feb 01 2021 11:59PM.

Kind regards,

Gabriel Gasque, Ph.D.,

Senior Editor

PLOS Biology

---

## [Decision Letter · Decision Letter 1]

2 Mar 2021

Dear Dr Gillis,

Thank you very much for submitting your manuscript "Assessing the replicability of spatial gene expression using atlas data from the adult mouse brain" for consideration as a Research Article at PLOS Biology. Your manuscript has been evaluated by the PLOS Biology editors, by an Academic Editor with relevant expertise, and by two independent reviewers.

In light of the reviews (below), we will not be able to accept the current version of the manuscript, but we would welcome re-submission of a much-revised version that takes into account the reviewers' comments. We cannot make any decision about publication until we have seen the revised manuscript and your response to the reviewers' comments. Your revised manuscript is also likely to be sent for further evaluation by the reviewers.

We expect to receive your revised manuscript within 3 months. 

**IMPORTANT - SUBMITTING YOUR REVISION**

Your revisions should address the specific points made by each reviewer. As you will see, the reviewers think the work is novel and important. However, they have raised concerns that you would need to address with clarifications, more discussion, and a significant re-write to focus on the neurobiological significance of the work. The reviewers also question some of your analyses, and you might need to provide additional ones to satisfy the referees. 

Editorially, we are interested in pursuing your study as a Methods and Resources Article and not as a Research Article. Please change the article type when re-submitting. 

Please submit the following files along with your revised manuscript:

*Re-submission Checklist*

*Published Peer Review*

*PLOS Data Policy*

*Blot and Gel Data Policy*

Sincerely,

Gabriel Gasque, Ph.D.,

Senior Editor,

ggasque@plos.org,

PLOS Biology

REVIEWS:

Reviewer #1: This is a well-written paper on a timely topic. Spatial gene expression analyses are currently becoming increasingly important, and the question of their replicability/generalizability is a crucial one.

However, I have a couple of minor and major concerns:

1. It would be good to clearly define the term "benchmarking".

2. The role of batch effects could be discussed in more details, both within datasets and across datasets. In other settings, it has been shown that correction for batch effects may improve cross-study classification. Could it also be the case here? 

3. p5: It would be helpful to summarize the format of the data (how many rows, columns, and what they correspond to).

4. As far as I can understand, each dataset corresponds to one brain. It would be good to clearly state that, or to clarify the text if I am wrong.

5. Did the authors consider using dedicated multi-class classification methods rather than considering pairwise comparisons?

6. The regularization parameter in lasso is usually called lambda. 

7. Two PCs capturing as much as 80% of the variance seems suspicious to me. Is it possible that large batch effects are an issue for this dataset, i.e. that data different brain regions have been collected/preprocessed in different ways, which would explain the very good classification accuracy? 

8. p11: "we picked 100 randomly seeded feature sets": How large are these feature sets?

9. p16: I don't understand how the authors can apply simple linear regression to this dataset. Are the data not high-dimensional? 

10. p22: The authors used 2-fold CV. That is fine, but I am wondering whether the results are stable. Would they have been the same with another random split into training and test data?

11. p22: "For within ST training and testing,...": It is a bit confusing to repeat that, since it has already been said at the beginning of the paragraph (reading this I got confused and wondered whether the sentence before was only about ST or general).

12. The cross-dataset analysis is based on the 14,299 overlapping genes, i.e. on a subset of genes. It would be interesting to see the within dataset performance obtained using only these genes, to check that the worse performance in cross-dataset learning is not due to important genes being absent.

13. p23: "The AUROC is typically...": This sentence is already about the *area under* the ROC curve, so the following sentence is redundant. Note that the "analytical AUROC" (as the authors call it) is also an estimate of the true unknown AUROC. 

14. p23: The AUROC formula given by the authors is not a standard one. Please give a standard one or specify a reference.

15. p23: Why were AUROCs of 0 removed from dowstream reporting? This may introduce a bias. What about values between 0 and 0.5?

16. p23: "We use a threshold that includes all instances of the positive label". I don't understand that. How do the authors choose with class of the pair is the "positive one"? Or do they perform two analyses successively for each pair?

17. p23: "Note, that for while": typo

18. p23: What is the rationale behing the statement at the end of p23/beginning of p24?

19. p24: It is bad practice to set the penalty parameter to an arbitrary value. This parameter has a huge importance on the performance of the model. It should be chosen using cross-validation. Otherwise, one cannot garantee that, for example, the better accuracy for ABA is due to 0.1 being more appropriate for this dataset than for ST.

20. p24: Please define the notation ||.||1.

21. p24: "PCA was applied [...] to each leaf brain". I don't understand that. Did the authors really repeat the PCA analyses for each leaf brain successively? Does that make sense? If yes, how did they compute the reported proportion of explained variability?

Reviewer #2: This work compares the identifiability of anatomic structures by gene expression comparing in situ hybridization data from the Allen Mouse Brain Atlas to modern spatial transcriptomic data capable of labeling cells expressing specific genes. The study is in principle well-conceived in that it very useful to compare cell type specific expression patterns and their anatomic identifiability between these modalities. The potential for one to inform the other is both important and interesting. This study in this way is very timely in comparing a widely used standard data set with more modern powerful expression profiling to understand key cellular co-expression relationships. The success of this method, in particular using the expression energy of the 200 cubic micron grid data is both interesting and surprising. In a sense this grid data is highly smoothed and a very coarse representation of the cell specific expression in the ABA. Yet apparently it is rich enough to classify regions. This is an important result, and I would be this forward in the writing. 

In general, the abstract and much of the paper could be written to elucidate the neurobiological significance of the results. The paper is written form a very informatics centric approach and reading it suggests the work of a strong informatics analyst and the senior authors on the paper might add more context and neurobiological significance of the work. The paper would be considerably improved by stressing the implications and utility of the work rather than details of performance classification. 

As an immediate example, starting the article's abstract with the sentence chosen is not very powerful with respect to the study. One of the value propositions of spatial transcriptomics is far from simply being a means to link data type, but rather to understand cell type specific expression in situ and its spatial distribution. This should be the spirit of opening the article in my opinion and not as a data link mechanism. 

Lines 100-107. The enumeration of these opportunities. I would state more directly option 5 as what is being done with an explanatory justification of why.

The methodology of the approach needs to be intuitively described early in the results section. That is, how are being performing the anatomic label recognition from the regularized fit? Not a technical description which can be delayed until the methods section but the reader needs to understand the computational nature of the approach before describing results. Specifically, line 161, what does it mean to classify a leaf ARA structure?

Line 195-196. This low dimensional characterization for small numbers of PC's accounting for substantial variance has also been observed by others and as the authors not likely arrives to highly spatially correlated expression patterns. However, these major axes of variance describe only the highest level of variation and must be seen in contrast to the result you demonstrate that leaf level structures in the ARA ontology can be identified. The conclusion I draw is that the PC approach offers little in the way of biological interpretation distinguishing small anatomic structures.

In a sense the ability of the ABA data to predict anatomic labels better than the spatial transcriptomic data may not be surprising. The spatial data is operating at a cell type specific label and may not have sufficient power to delineate anatomic at this coarser level. At the ABA level as the authors note this expression identifiability is helped by considerable auto-correlation in the spatial signal and smoothing due to partial volume effects of the smoothed grid data. The authors may offer potential insights into these differences and with respect to the anatomic interpretability of spatial transcriptomic data.

I see as somewhat problematic the use of the sagittal ABA data in this context, here the registration and hence the gridded data is less accurate and voxel level. This may not add that much to the analysis, in particular as the coronal data set was chosen to contain many of the most likely differentially expression genes and these will be the differentiating ones in the LASSO regression analysis.

All in all this is an interesting paper that could be more impactful with a more implication focused delivery while relegating details of prediction tradeoffs and performance relegated to supplementary materials.

---

## [Decision Letter · Decision Letter 2]

29 Jun 2021

Dear Dr Gillis,

On behalf of my colleagues and the Academic Editor, Franck Polleux, I am pleased to say that we can in principle offer to publish your Methods and Resources "Assessing the replicability of spatial gene expression using atlas data from the adult mouse brain" in PLOS Biology. Your manuscript was re-evaluated by the original reviewers, who feel the revised manuscript has satisfactorily addressed their concerns (full comments below). 

Before final acceptance and publication, we will need you to address any remaining formatting and reporting issues. These will be detailed in an email that will follow this letter and that you will usually receive within 2-3 business days, during which time no action is required from you. Please note that we will not be able to formally accept your manuscript and schedule it for publication until you have made the required changes.

PRESS

Sincerely, 

Luke Smith (on behalf of Gabriel Gasque) 

Associate Editor 

PLOS Biology

lsmith@plos.org

REVIEWER COMMENTS:

Reviewer #1: The authors correctly addressed my concerns.

Reviewer #2 (Michael Hawrylycz): The authors have provided a comprehensive revision and addressed all of the considerations that I raised. A significant amount of new supporting material has been generated and I can recommend publication.